# Exposure to hand-held vibrating tools and biomarkers of nerve injury in plasma: a population-based, observational study

Malin Zimmerman ,[1,2] Peter Nilsson ,[3] Lars B. Dahlin [1,4]

¹Department of Translational Medicine, Lund University, Malmö, Sweden
²Department of Orthopedics, Helsingborg's Hospital, Helsingborg, Sweden
³Department of Clinical Sciences, Lund University, Malmö, Sweden
⁴Department of Biomedical and Clinical Sciences, Linköping University, Linköping, Sweden

**Correspondence to**
Dr Lars B. Dahlin;
lars.dahlin@med.lu.se

## ABSTRACT

**Objectives** To analyse potential biomarkers for vibration-induced nerve damage in a population-based, observational study.

**Design** Prospective cohort study.

**Setting** Malmö Diet Cancer Study (MDCS), Malmö, Sweden.

**Participants** In a subcohort of 3898 individuals (recruited 1991–1996) from MDCS (baseline examination in 28 449 individuals; collection of fasting blood samples in a cardiovascular subcohort of MDCS of 5540 subjects), neuropathy-relevant plasma biomarkers were analysed during follow-up after filling out questionnaires, including a question whether work involved hand-held vibrating tools, graded as 'not at all', 'some' or 'much'.

**Primary outcome measures** The neuropathy-relevant plasma biomarkers vascular endothelial growth factor (VEGF)-A, VEGF-D, VEGF receptor 2, galanin, galectin-3, HSP27, ß-nerve growth factor, caspase-3, caspase-8, transforming growth factor-$\alpha$ and tumour necrosis factor were analysed. Data were analysed by conventional statistics (Kruskal-Wallis test; post hoc test Mann-Whitney U test; Bonferroni correction for multiple testing) and in a subanalysis for galanin using two linear regression models (unadjusted and adjusted).

**Results** Among participants, 3361 of 3898 (86%) reported no work with hand-held vibrating tools, 351 of 3898 (9%) reported some and 186 of 3898 (5%) much work. There were more men and smokers in vibration-exposed groups. Galanin levels were higher after much vibration exposure (arbitrary units 5.16±0.71) compared with no vibration exposure (5.01±0.76; p=0.015) with no other observed differences.

**Conclusions** Higher plasma levels of galanin, possibly related to magnitude, frequency, acceleration and duration, as well as to severity of symptoms of vibration exposure, may be found in individuals working with hand-held vibrating tools.

## INTRODUCTION

Hand-arm vibration syndrome (HAVS) is a common condition that may induce injuries to a variety of the tissues in the upper extremity, particularly affecting the peripheral nervous and musculoskeletal systems. Approximately 14% of all employed men and

## STRENGTHS AND LIMITATIONS OF THIS STUDY

⇒ Based on data from the Malmö Diet Cancer Study, neuropathy-relevant plasma biomarkers could be analysed in a subcohort of 3898 individuals.
⇒ The neuropathy-relevant biomarkers were related to replies from a questionnaire if the participant's work involved exposure to hand-held vibrating tools and graded as 'not at all', 'some' or 'much'.
⇒ A wide range of biomarkers were analysed, consisting of vascular endothelial growth factor (VEGF)-A, VEGF-D, VEGF receptor 2, galanin, galectin-3, HSP27, ß-nerve growth factor, caspase-3, caspase-8, transforming growth factor-$\alpha$ and tumour necrosis factor, with focus on galanin, a relevant protein involved in several biological processes, including nociception.
⇒ The generalisation of the findings of galanin has in the future to be related to extent of exposure, that is, various types of magnitude, frequency, acceleration and duration, in individuals working with hand-held vibrating tools.

3% of women in Sweden are exposed to vibrations from hand-held tools for at least 25% of their working time.[1] The amount of induced injury to the individual's extremity depends on the magnitude, frequency and acceleration of the vibration.[2] Vibration exposure may induce the well-known white finger symptoms, but paresthesia, numbness and impaired sensation in the affected extremity, based on an underlying nerve injury, are probably more common.[3] To confirm a proper HAVS diagnosis, the treating physician must determine a history of vibration exposure, rule out other explaining conditions and perform the different clinical assessment methods to detect an impaired function.[4] However, there are no specific simple reported biomarkers that can be measured in the blood to support, for example, signs of a neurological injury. The prevalence of vibration-induced neuropathy in HAVS may extend from 7% to 79% (average 39%).[3]

Such symptoms may be caused by structural changes, for example, extensive axonal degeneration, as observed in biopsies from the posterior interosseous nerve just proximal to the wrist in men exposed to vibrations.[5] Even data from skin biopsies of fingers in vibration-exposed workers indicate that there is a reduction in content of calcitonin gene-related peptide as well as intraepidermal nerve fibre density observed as staining for PGP-9.5 in the biopsies.[6] The structural changes in the peripheral nerve may also induce an increased susceptibility for peripheral nerve compression syndromes, like carpal tunnel syndrome (CTS),[7] in accordance with the relation between diabetic neuropathy and CTS.[8][9] The aim of the present observational study was to examine several nerve-related biomarkers, as a sign of neuropathy that can be measured in plasma in a cohort of subjects who were exposed to hand-held vibrating tools.

## METHODS

### Study population

The cohort Malmö Diet Cancer Study (MDCS)[10] consists of 30 446 individuals from the southern part of Sweden. The individuals, living in the city of Malmö and born in 1926–1945, were invited to participate in the study during 1991–1996. The individuals underwent a health examination and assessment of laboratory tests, and filled in a large number of questionnaires regarding cardiovascular risk factors as previously described in detail.[10] The rate of attendance was approximately 41%, of which 60% of the individuals were women. The characteristics of the cohort and the inclusion criteria for MDCS have previously been described.[10–14] From the large cohort of 28 449 individuals (11 246 men and 17 203 women) who underwent adequate baseline examination, 6103 individuals were invited to the specific cardiovascular subcohort of the MDCS, where 5540 subjects accepted inclusion for collection of fasting blood samples. The details of the laboratory analysis were recently presented.[15][16] In short, blood samples were analysed using Olink Proseek Multiplex proximity extension assay, which uses two antibodies that bind to the target protein.[16] In the final subcohort, 4865 samples were sent for analysis and after internal quality control, 4741 individuals remained for evaluation of biomarker panels. In 843 individuals, data on vibration exposure were missing, resulting in 3898 included individuals for the present study.

To analyse vibration exposure, the following question was included in participants' questionnaires: 'Does your work involve working with hand-held vibrating tools?' Participants graded their exposure to hand-held vibrating tools as 'not at all', 'some' and 'much', but the duration of exposure or the type of equipment was not defined or reported by the participants. During follow-up, the specific laboratory tests as described were performed and the following neuropathy-relevant biomarkers were selected: vascular endothelial growth factor (VEGF)-A, VEGF-D, VEGF receptor 2, galanin, galectin-3 (Gal3),

HSP27, ß-nerve growth factor (NGF), caspase-3, caspase-8, transforming growth factor (TGF)-α and tumour necrosis factor (TNF). Three of these biomarkers have been used in a recent study related to development of the two nerve compression disorders: CTS and ulnar nerve entrapment.[15]

### Statistical analysis

Nominal data are presented as number (%) and were compared using the $X^2$ test. The biomarkers are presented as mean±SD and are expressed as arbitrary units. The Kruskal-Wallis test was used to analyse potential differences between groups with a Bonferroni correction due to the number of biomarkers (11 biomarkers; that is, requiring p<0.004). In subsequent Mann-Whitney pairwise comparisons, p values were also adjusted by the Bonferroni correction for multiple testing. Linear regression models were used to study the effect on vibration exposure on levels of galanin (see the Results section on reason why galanin was selected). Two models were used, one unadjusted (model 1) and one adjusted for age at baseline, sex, diabetes and smoking. A p value of <0.05 was considered significant. IBM SPSS Statistics for Mac V.28.0 was used for the statistical analysis.

### Patient and public involvement

No patients, politicians or members of the public were involved in the development of the research question and the design of the present study. The results of the research, after being published in the scientific journal, will be disseminated in a variety of ways. One way will be information to physicians and surgeons working with patients with HAVS in everyday practice and through the healthcare organisations connected to small and large companies. Furthermore, the results will be spread through lectures to other colleagues and students in the healthcare sector and to any patient organisations or insurance companies with interest in HAVS.

## RESULTS

In the total number of individuals, 1603 of 3898 (41%) were men and 2295 of 3898 (59%) were women with a mean age of 57±6 years. At baseline, 3361 of 3898 (86%) reported that they did not at all work with hand-held vibrating tools, while 351 of 3898 (9%) and 186 of 3898 (5%) reported that their work involved some or much work with hand-held vibrating tools, respectively. There were more men and more smokers in the groups who were exposed to vibrations. The baseline characteristics of the individuals in this study are presented in table 1.

### Biomarkers

Observed differences in biomarker plasma levels were small (table 1). Galanin levels were higher in the group reporting much vibration exposure compared with the group without vibration exposure (table 1). No other differences among the other analysed biomarkers were

**Table 1** Biomarkers in plasma in workers reporting no, some or much vibration exposure

| | 'Does your work involve working with hand-held vibrating tools?' | | | |
| | Not at all (n=3361) | Some (n=351) | Much (n=186) | P value |
|---|---|---|---|---|
| Men n (%) | 1215 (36) | 236 (67) | 152 (82) | <0.001 |
| Age (years) | 57±6 | 58±6 | 57±6 | 0.23 |
| Smoking n (%) | 860 (26) | 107 (30) | 63 (34) | 0.009 |
| Diabetes n (%) | 661 (20) | 74 (21) | 32 (17) | 0.56 |
| VEGF-A (n=3898) | 10.0±0.49 | 10.0±0.44 | 10.1±0.45 | 0.25 |
| VEGF-D (n=3896) | 6.74±0.48 | 6.70±0.48 | 6.74±0.57 | 0.27 |
| VEGFR2 (n=3788) | 6.86±0.39 | 6.84±0.39 | 6.81±0.37 | 0.27 |
| Galanin (n=3898) | 5.01±0.76 | 5.08±0.77 | 5.16±0.71 | 0.003* |
| Galectin-3 (n=3898) | 5.15±0.42 | 5.10±0.44 | 5.16±0.37 | 0.12 |
| HSP27 (n=3897) | 4.76±0.84 | 4.79±0.80 | 4.84±0.78 | 0.46 |
| ß-NGF (n=403) | 1.18±0.39 | 1.36±0.67 | 1.12±0.25 | 0.38 |
| Caspase-3 (n=3718) | 10.74±0.95 | 10.80±0.90 | 10.79±0.93 | 0.52 |
| Caspase-8 (n=3865) | 1.58±0.69 | 1.64±0.72 | 1.70±0.74 | 0.10 |
| TGF-α (n=3786) | 1.39±0.34 | 1.38±0.33 | 1.36±0.32 | 0.31 |
| TNF (n=128) | 1.33±1.68 | 1.20±1.95 | 2.18±3.1 | 0.79 |

Biomarkers presented as mean level of arbitrary units±SD. Statistical analysis based on $X^2$ (sex, smoking and diabetes) or Kruskal-Wallis test, requiring p<0.004 (11 biomarkers).
*In subsequent Mann-Whitney pairwise comparisons, the statistically significant difference was found between 'not at all' and 'much' with a p value of 0.015 adjusted by the Bonferroni correction for multiple tests.
NGF, nerve growth factor; TGF, transforming growth factor; TNF, tumour necrosis factor; VEGF, vascular endothelial growth factor; VEGFR2, VEGF receptor 2.

observed. When further analysing galanin levels, being in the group who reported high vibration exposure predicted higher levels of galanin in the unadjusted model (table 2). When adjusting for age, sex, smoking and diabetes, no significant predictions could be made in the regression model (table 2).

## DISCUSSION

The present population-based, observational study indicates that individuals with exposure to hand-held vibrating tools in a large cohort of around 30 000 patients, where levels of several biomarkers related to nerves, and possible vibration-induced neuropathy, were available in around 3898 individuals with also information about exposure to hand-held vibrating tools, showed increased levels of plasma galanin. In contrast, other relevant nerve-related biomarkers, including the neuroprotective substance HSP27, which is relevant to diabetic neuropathy[17 18] and nephropathy,[19] did not show any statistically significant alterations among the vibration-exposed individuals.

Galanin is involved in several biological processes,[16 20] including nociceptive sensory processing in the spinal cord. Galanin and neuropeptide Y are consistently upregulated on both gene and protein levels in dorsal root ganglia (DRG) in rodents with a potential relation to pain after a severe nerve injury,[21–23] a symptom which may

**Table 2** Linear regression models of galanin levels in plasma in workers reporting no, some or much vibration exposure

| | Model 1 | Model 2 |
|---|---|---|
| No vibration exposure | Reference | Reference |
| Some vibration exposure | 0.06 (–0.02 to 0.15) | –0.04 (–0.12 to 0.04) |
| Much vibration exposure | 0.14 (0.03 to 0.26) | –0.02 (–0.13 to 0.09) |
| Age | | –0.01 (–0.01 to –0.01) |
| Sex (female reference) | | 0.41 (0.36 to 0.46) |
| Smoking | | –0.32 (–0.37 to –0.27) |
| Diabetes | | –0.13 (–0.19 to –0.08) |

Unadjusted B coefficients with 95% CI. Model 1 unadjusted and model 2 adjusted for age, sex, smoking and diabetes.

also be present among individuals with HAVS.[24 25] Experimental studies in rats also indicate that a sciatic nerve compression, which is a less pronounced injury, increases the number of galanin-stained sensory neurons in DRG compared with uninjured and contralateral sensory neurons.[26] Furthermore, a complete nerve transection is more efficient than a partial nerve transection or a slight compression injury as an inducer of galanin expression in DRG sensory neurons.[26] Such injuries seem to induce an upregulation of galanin in mainly small diameter sensory neurons and to a lesser degree in larger neurons.[26] Thus, the extent of induction of galanin expression is related to the severity of nerve injury.[27] Interestingly, the induction of galanin in small diameter neurons in the DRG, which are related to smaller diameter nerve fibres in the sciatic nerve, is notable, since small diameter nerve fibres are reported to be affected by experimental vibration exposure in rat paws.[28] The structural changes seem to occur close to the source of vibration exposure and not more proximally.[28] Whether the present increase in galanin is due to upregulation of galanin in small or large diameter nerve fibres is not possible to analyse in our cohort. However, our results indicate that a self-reported history of slight or moderate exposure to hand-held vibrating tools is sufficient to provoke a nerve injury that results in increased, and persistent, plasma levels of galanin. On the other hand, it is not surprising that, for example, β-NGF did not show any increased plasma levels, since such a substance is only transiently upregulated after a severe nerve injury (ie, nerve transection)[29] and vibration exposure may hence not be reflected by any change in β-NGF in plasma, which is probably also applicable for some of the other biomarkers. Pro-inflammatory (eg, TNF) and anti-inflammatory (TGF-ß) cytokines are expressed after Wallerian degeneration, where the former promotes recruitment of macrophages, and the latter are expressed after macrophage recruitment with a task to attenuate inflammation.[30] The present vibration exposure was not severe enough to induce a nerve injury resulting in higher levels of such cytokines among the participants with much vibration exposure. Another relevant protein is Gal3, which is considered to modulate cell to extracellular matrix interactions, but also as a mediator of signal transduction event on the cell surface as well as a variety of extracellular processes.[31] This multicellular protein was selected to be analysed in relation to vibration exposure since one suspect it may be increased in view of described structural changes in affected nerves. However, the plasma levels were not different between the participants with and without vibration exposure.

Thus, we did not find any increased plasma levels of the other measured biomarkers. The caspases indicate apoptosis of, for example, Schwann cells that occur in a peripheral nerve after injury in experimental systems.[32] A more severe experimental nerve injury may also induce increased upregulation of HSP27 locally in the peripheral nerve as well as in sensory neurons.[33] Furthermore, a generalised neuropathy, such as diabetic neuropathy, increases plasma levels of HSP27, with an association between plasma levels and protection of nerve function.[17 18] Other neuropathies, including diabetic neuropathy, also increase a variety of plasma proteins that indicate affection and injury to the nerve fibres, that is, axons and Schwann cells, as well as to vascular components.[34–36] However, we did not find any increased plasma levels of these biomarkers in the present cohort. There are also reports that there is a significantly altered expression of insulin growth factor-1 immunoreactivity in tendons and in nerves after experimental vibration exposure.[37 38] VEGF and its receptors may also be of relevance since they have been implicated in treatment of nerve regeneration[39] and in diabetic neuropathy.[40] This factor, initially defined as being angiogenetic, is described as relevant in the cross-talk between peripheral nerve components and blood vessels,[41] again being of potential interest in HAVS due to the vascular and neural component in the syndrome. However, even the participants' opinion of much vibration exposure was not sufficient to increase the plasma levels. Thus, a number of potentially relevant biomarkers were selected and analysed, but were not found to be high in the plasma. Recently, low molecular organic biomarkers were analysed in blood samples in vibration-exposed workers using metabolomics, showing differences in those with vibration-induced white fingers.[42] However, these metabolites were not possible to analyse in the present cohort. The interpretation of the present results is that vibration exposure is not sufficient to constantly increase any upregulation of a variety of biomarkers, except galanin, in the peripheral nerve that is mirrored in the plasma levels.

### Strengths and limitations

The percentage of participants who reported exposure to hand-held vibrating tools was 14%, which is slightly lower than official data in Sweden.[43] Thus, it is reported that 19% of the population (all types of professions) work with hand-held machines at least 25% of the working time 2017–2019 (interpreted to be relatively unchanged during the last decades) with a difference between men (25%) and women (12%).[44] Part of the difference may be due to the specific question asked in the present study (*Does your work involve working with hand-held vibrating tools?*), which could be interpreted by the participants as both a limited as well as an extensive exposure. In addition, since the question was referring to hand-held 'vibrating tools', we anticipate that the vibrations were of certain frequencies and most probably not very low frequencies. We had no information about disease or severity stages, for example, the Stockholm scales, which is a limitation. A further limitation of the study is that we had no detailed data on the exact level of vibration exposure for each participant, such as cumulative exposure (ie, magnitude, frequency and acceleration), which would have enabled more detailed analysis with prediction of risk for a higher level of galanin in relation to cumulative exposure. A specified analysis of cumulative exposure requires detailed analysis

of each participant with preferably measurements at the working place and related to their profession. However, the strengths include the well-defined cohort and the large study sample.

## CONCLUSIONS

We conclude that higher levels of plasma galanin may be found in individuals reporting work with hand-held vibrating tools. Future research should investigate whether galanin might be useful as a biomarker in HAVS in relation to vibration exposure at various types of magnitude, frequency, acceleration and duration as well as to severity of symptoms.

**Acknowledgements** The authors are very grateful to all the participants in MDCS whose participation enabled this work. They are also grateful to Anders Dahlin for his help with data extraction and Tina Folker for her administrative assistance.

**Contributors** A significant contribution was made by all authors stated in this article. MZ, PN and LD designed the study. The first draft was made by MZ and LD, and all authors contributed to the interpretation of the data. Finally, all authors reviewed and accepted the final version before publishing. LBD is the guarantor of the work.

**Funding** This work was supported by grants from Region Skåne, Lund University, Kockska stiftelsen, Stig and Ragna Gorthon Foundation, the Swedish Diabetes Foundation (DIA2020-492), the Swedish Research Council (2022-01942), the Regional Agreement on Medical Training and Clinical Research (ALF) between Region Skåne and Lund University and Magnus Bergvall Foundation (2020-03612).

**Disclaimer** The sponsors had no role in the design of the study, collection, analysis, and interpretation of data, writing of the report or submission of the paper.

**Competing interests** None declared.

**Patient and public involvement** Patients and/or the public were not involved in the design, or conduct, or reporting, or dissemination plans of this research.

**Patient consent for publication** Not required.

**Ethics approval** This project has been approved by the Regional Ethical Review Board in Lund, Sweden (51-90, 2009-633, 2011-356, 2014-742, 2015-686, 2015-754, 2016-479) and by the Swedish Ethical Review Authority (2019-01439). All participants signed informed consent to participate in the cohort.

**Provenance and peer review** Not commissioned; externally peer reviewed.

**Data availability statement** Data may be obtained from a third party and are not publicly available. Public access to data is restricted by the Swedish Authorities (Public Access to Information and Secrecy Act; https://government.se/information-material/2009/09/public-access-toinformation-andsecrecy-act/), but data can be made available for researchers after a special review that includes approval of the research project by both the national Swedish Ethical Review Authority and the authorities' data safety committees.

**ORCID iDs**
Malin Zimmerman http://orcid.org/0000-0002-9925-3838
Peter Nilsson http://orcid.org/0000-0002-5652-8459
Lars B. Dahlin http://orcid.org/0000-0003-1334-3099

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
