## [Reviewer comments · BMJ Open]

ARTICLE DETAILS

TITLE (PROVISIONAL)	Exposure to vibrating hand-held tools and biomarkers of nerve injury in plasma - a population-based, observational study
AUTHORS	Zimmerman, Malin; Nilsson, Peter; Dahlin, Lars

VERSION 1 – REVIEW

REVIEWER	Foit, Niels McGill University
REVIEW RETURNED	06-Jan-2023

GENERAL COMMENTS	The authors present a manuscript on the identification of potential biomarkers for work related vibration neuropathy. This study is clearly warranted due to its implications for the working population. Moreover, availability of a biomarker to identify such injury is still lacking and currently relies on a combination of history, electrophysiology and nerve conduction studies. The study is accurate and the results are presented clearly. Nevertheless, I was surprised to see so few participants who actually used vibrating tools - only approx 500. in this relatively small cohort, galanin levels were elevated. while these results clearly indicate a cause-symptom relationship, the authors do not elaborate or discriminate between e.g. duration of vibration exposure and biomarker levels. moreover, a prediction model should be specified, which takes exposure, intensity etc. into consideration.
---

REVIEWER	Mondal, Banashree Institute of Neurosciences Kolkata, neurology
REVIEW RETURNED	14-Jan-2023

GENERAL COMMENTS	1. A little detail of the vibration exposure can be explained.2. In the method section if you can mention the year of exposure of these patients3. What type of exposure did these patients experience?4. Is there any disease stage or severity index?5. What were the symptoms of the HAVS?6. Is it possible to correlate the symptoms with biomarkers?
--

VERSION 1 – AUTHOR RESPONSE

Reviewer: 1

Dr. Niels Foit, McGill University

Comments to the Author:

The authors present a manuscript on the identification of potential biomarkers for work related vibration neuropathy. This study is clearly warranted due to its implications for the working population. Moreover, availability of a biomarker to identify such injury is still lacking and currently relies on a combination of history, electrophysiology and nerve conduction studies. The study is accurate and the results are presented clearly. Nevertheless, I was surprised to see so few participants who actually used vibrating tools - only approx 500. In this relatively small cohort, galanin levels were elevated. While these results clearly indicate a cause-symptom relationship, the authors do not elaborate or discriminate between e.g. duration of vibration exposure and biomarker levels. Moreover, a prediction model should be specified, which takes exposure, intensity etc. into consideration.

Reply: Thank you for your most adequate points.

We agree about the relatively low number of participants working with vibrating hand-held tools (i.e. 537/3898; 14%). In Sweden it has been anticipated that around 19% of the working population are exposed to vibrating tools at least 25% of the working time during work (difference between men and women). A comment about the difference with reference is included in Discussion.

We also agree that it would be most interesting to add data, with statistical sub-analysis, about duration and extent of vibration exposure. However, when the present register was created, with the subset of the cardiovascular cohort, it was not a specific focus to perform a detailed analysis of vibration levels and duration. The present study rather indicates that for example galanin may be one important factor that should be included in future studies. A further clarification is added in Discussion.

Reviewer: 2

Ms. Banashree Mondal, Institute of Neurosciences Kolkata

Comments to the Author:

1. A little detail of the vibration exposure can be explained.

Reply: Please, see comment above.

2. In the method section if you can mention the year of exposure of these patients

Reply: We can only define the vibration exposure based on the specific question, indicating if the participant worked with such tools at the time of replying to the questionnaire, but not for how long. Comment is added in Discussion.

3. What type of exposure did these patients experience?

Reply: We can only relate our data to the question asked in the questionnaire. i.e., “vibrating hand-held tools”, indicating that it is not low or high, or both, frequency vibrations. Comment added in Discussion.

4. Is there any disease stage or severity index?

Reply: Again, since we have no details about the exposure, we cannot grade the participants in any disease or symptom grades, such as the Stockholm scale. Comment added in limitation.

5. What were the symptoms of the HAVS?

Reply: Please, see comment in point 4.

6. Is it possible to correlate the symptoms with biomarkers?

Reply: Since we had no information about symptoms, we could not make any correlations. Presently, the relation between levels of biomarkers to various symptoms is a most interesting research area and will be further investigated in the near future. In addition, we have added some references concerning plasma proteins in neuropathies as a comparison.

We do hope that these changes and answers are sufficient and that you now will consider the manuscript suitable for publication.

On behalf of the authors

Lars B. Dahlin